# Carriage rates and antimicrobial sensitivity of pneumococci in the upper respiratory tract of children less than ten years old, in a north Indian rural community

**Sambuddha Kumar[1], Debjani Ram Purakayastha[2], Arti Kapil[1], Siddhartha Saha[3], Fatimah S. Dawood[3], Bimal Kumar Das[1]\*, Ritvik Amarchand[2], Rakesh Kumar[2], Kathryn E. Lafond[3], Seema Jain[3], Anand Krishnan[2]**

1 Department of Microbiology, All India Institute of Medical Sciences, New Delhi, India, 2 Center for Community Medicine, All India Institute of Medical Sciences, New Delhi, India, 3 Centers for Disease Control and Prevention, Atlanta, Georgia, United States of America

\* tezpur.bimal@gmail.com

**Data Availability Statement:** All relevant data are within the paper and its Supporting information files.

## Abstract

Pneumococcal carriage studies are important for vaccine introduction and treatment strategies. Pneumococcal carriage rates estimated in this cohort study among children in a rural community of northern India. Between August 2012 and August 2014, trained nurses made weekly home visits to screen enrolled children aged <10 years for acute upper or lower respiratory infections (AURI/ALRI) in Ballabgarh, Haryana. Nasal swab from infants aged <1year and throat swab from children aged ≥1 year were collected. All specimens were cultured for pneumococci; isolates were serotyped and subjected to antimicrobial susceptibility testing. During the study period, 4348 nasal/throat swabs collected from children with clinical features of ARI (836 ALRI, 2492 AURI) and from 1020 asymptomatic children. Overall pneumococcal carriage was 5.1%, the highest carriage rate among children <1 year of age (22.6%). The detection rates were higher among children with ARI (5.6%; 95% CI: 4.8–6.4) than asymptomatic children (3.3%; 95% CI: 2.3–4.6). Among 220 pneumococcal isolates, 42 diverse serotypes were identified, with 6B/C (8.6%), 19A (7.2%), 19F (6.8%), 23F (6.4%), 35A/B/C (6.4%), 15B (5%), 14 (4.5%) and 11A/C/D (3.2%) accounting for 50%. Forty-five percent of the serotypes identified are included in the current formulation of 13-valent pneumococcal conjugate vaccine. Ninety-six percent of isolates were resistant to co-trimoxazole, 9% were resistant to erythromycin, and 10% had intermediate resistance to penicillin with minimum inhibitory concentration ranges (0.125 to 1.5 µg/ml). Pneumococcal detection was relatively low among children in our study community but demonstrated a diverse range of serotypes and half of these serotypes would be covered by the current formulation of 13-valent pneumococcal vaccine.

**Funding:** This study was funded under the cooperative agreement between the All India Institute of Medical Sciences, New Delhi, India and Center for Disease Control and Prevention, Atlanta, USA (U01-IP000492). Sambuddha Kumar, Debjani Ram Purakayastha, Arti Kapil, Siddhartha Saha, Fatimah S. Dawood, Bimal Kumar Das, Ritvik Amarchand, Rakesh Kumar, Kathryn E. Lafond, Anand Krishnan were all involved in the planning of the study and development of the study protocols and tools. Debjani Ram Purakayastha, Rakesh Kumar, Ritvik Amarchand coordinated community-based surveillance activities. Sambuddha Kumar, Bimal Kumar Das and Arti Kapil performed the bacteriological studies. Sambuddha Kumar, Rakesh Kumar, Ritvik Amarchand and Siddhartha Saha analyzed the data. Sambuddha Kumar, Arti Kapil drafted the manuscript. Siddhartha Saha, Seema Jain, Kathryn E. Lafond, Fatimah S. Dawood, Anand Krishnan provided overall supervision for the project activities. All the authors have read and reviewed the manuscript.

**Competing interests:** No competing interest.

# Introduction

According to the World Health Organization (WHO) and Maternal and Child Epidemiology Estimates Group (MCEE), in 2015, India had the highest number of under-five deaths due to acute respiratory infection (ARI) in the world [1, 2]. Although pneumococcal disease is vaccine preventable, *Streptococcus pneumoniae* remains a major cause of acute lower respiratory infection (ALRI) in children, causing an estimated 294,000 under-five deaths [3]. Pneumococcal disease is a major public health problem in India. Almost a quarter of global pneumococcal cases and deaths occur in India [3, 4].

India introduced pneumococcal vaccine in selected high burden states in 2017 with plans to roll out across the country to reduce the under-five mortality [5]. Pneumococcal carriage studies are useful for monitoring circulating serotypes and the proportion that are covered by available pneumococcal conjugate vaccines (PCV). Pre- and post-PCV introduction nasopharyngeal carriage prevalence and serotype distribution are also used as an indirect measure to assess vaccine impact [6]. The introduction of pneumococcal vaccine is expected to reduce the incidence of pneumococcal disease by reducing carriage rates of vaccine serotypes. However, in India, data on pneumococcal carriage are sparse. Few multi-site hospital-based studies from India found that serotypes 14, 1, 5, 19F and 6B were the most commonly identified serotypes in hospitalized children aged <5 years with invasive pneumococcal disease and the current PCV13 vaccine may provide serotype specific immunity against 74% of the serotypes causing invasive disease [7, 8].

Asian countries are unique with culture, society and somehow similar in lifestyle and overcrowded population of these regions is probably the major reasons for pneumococcal transmission. Pneumococcal distribution from Asian region countries, South Asian Pneumococcal Alliance (SAPNA) and Asian Strategic Alliance for Pneumococcal Disease Prevention (ASAP) reported circulating serotypes mostly involving in invasive pneumococcal disease [9]. Among neighbouring countries Pakistan had started expanded programme of immunization for PCV10 in 2012, with financial support of the Global Alliance for Vaccines and Immunisation (GAVI) and after that conjugate vaccine PCV10 included in Bangladesh and PCV13 in Nepal on 2015 [10].

As a part of the effort to reduce under-five deaths due to pneumonia, frontline health workers in various low- and middle-income countries have been trained to treat pneumonia with antimicrobials based on symptoms and signs using the Integrated Management of Childhood Illnesses (IMCI) strategy [11]. However, emergence of antimicrobial resistance (AMR) to commonly used antibiotics like co-trimoxazole hampers effective prevention and treatment of pneumonia [12, 13]. The data on AMR in India, especially for important pathogens such as *S. pneumoniae*, are scanty. Data from different regions are needed to guide antibiotic treatment strategies for children with pneumonia in India.

The present study was carried out to assess the carriage, serotypes and AMR of circulating pneumococcal isolates among children aged <10 years living in rural northern India, prior to pneumococcal vaccine introduction. As the concentration of pneumococcal carriage is known to increase during acute upper respiratory infection (AURI), and acute lower respiratory infection (ALRI) [14, 15], and also compared carriage rates among children with clinical features suggestive of AURI, ALRI and asymptomatic children.

# Methods

## Study population and specimen collection

From August 2012 to August 2014 children aged <10 years surveyed living in four villages of Ballabgarh block in Faridabad district of the northern state of Haryana, India. This work was

part of an ongoing cohort study on acute respiratory tract infections [16]. Briefly, trained study staff visited all the enrolled children at their homes weekly for the detection of ARI case defined as onset of at least one of the symptoms of cough, sore throat, rhinorrhea, shortness of breath or earache/ear discharge in past 7 days. Children with any of these symptoms and signs of tachypnea, stridor, chest in-drawing, lethargy, convulsions or unconsciousness were classified as acute lower respiratory infection (ALRI) using the age-appropriate Integrated Management of Neonatal and Childhood Illness (IMNCI) [11] and Integrated Management of Adolescent and Adult Illnesses (IMAI) guidelines [17]. Children with ARI without the signs and symptoms of ALRI were classified as having acute upper respiratory infection (AURI). Study nurses collected nasal swabs from infants aged <1year and throat swab from children aged 1–10 years. Nurses also collected swabs from similar number (ALRI) of conveniently selected asymptomatic children of similar age +/- 6 months from the same neighborhood with no respiratory symptoms in past 2 weeks. In addition, also collected specimens from a sample of AURI cases every day, as testing all samples was not possible because of limited resources. By rotation, on each workday of the week, one of the project nurses collected specimens from children with AURI detected in her designated surveillance area. As children were followed weekly, it is possible that a child may be swabbed more than once during the same episode if it lasts more than 7 days with worsening from AURI to ALRI. Swabs collected from children were immediately put into 15 ml falcon tube containing STGG (skimmed milk, tryptone, glucose, and glycerin) transport medium [18] and transported in an icebox per standard protocols to the bacteriology laboratory [19], at the All India Institute of Medical Sciences (AIIMS), New Delhi. In addition, separate nasal and throat swabs were also collected from each of these children with AURI/ALRI and the asymptomatic children for testing viral pathogens. [16].

## Laboratory methods

In the laboratory, specimens were vortexed to disperse organisms from the swab tip and cultured on Columbia agar plate with 5% sheep blood (bioMérieux, ™ France), and incubated overnight in a 5% $CO_2$ incubator at 37˚C. Pneumococcal isolates based on morphological structure of colony identified by α-hemolytic, microscopic observation, optochin susceptibility and bile solubility as recommended by the World Health Organization Pneumococcal Carriage Working Group [19]. Performed Neufeld's Quellung reaction for serotyping, using type-specific omni sera, pool sera, group sera, type sera and factor sera of Statens Serum Institut (SSI, Copenhagen, Denmark). Non-typeable isolates were further tested by PCR with fresh colonies of *S. pneumoniae*. DNA for PCR was extracted, using the QIAamp® DNA mini kit (Qiagen, Germany), according to the manufacturer's instructions. Molecular confirmation for *S. pneumoniae* isolates was performed using polymerase chain reaction (PCR) for the presence of the conserved sequence for autolysin encoding gene *lytA* [20]. Oligonucleotide primers used for 40 pneumococcal serotypes by conventional multiplex PCR method available on the CDC website (https://www.cdc.gov/streplab/downloads/pcr-oligonucleotide-primers.pdf). The condition for multiplex PCR was, 94˚C for 4 min, followed by 30 amplification cycles of 94˚C for 45 seconds, 54˚C for 45 seconds and 65˚C for 2 minutes and 30 seconds, while the annealing temperature for individual PCR reaction was at 60˚C [20]. The PCR products were analyzed by gel electrophoresis on 2% agarose gels.

Antimicrobial susceptibility testing conducted using antibiotic disc (Himedia Laboratories, Mumbai ™, India), by Kirby-Bauer disc diffusion method, as per the Clinical and Laboratory Standards Institute (CLSI) guidelines, for ceftriaxone, chloramphenicol, co-trimoxazole, erythromycin, levofloxacin, oxacillin, penicillin, vancomycin [21]. American Type Cell Collection (ATCC) 49619 of *S. pneumoniae* was used for a quality control test. Pneumococcal isolates that

were not susceptible to penicillin were further tested for minimum inhibitory concentration (bioMérieux, ™ France) using ETEST® strips [21].

### Data management and analysis

Considered specimens as unit of analysis, collected specimens from a child if s/he had a new case of ARI or had worsened from AURI to ALRI during the last seven days to detect new infections. Therefore, the nurses could have collected specimens from same child more than once; so analyzed the proportion of specimens positive for *S. pneumoniae*, counting multiple specimens from the same child as discrete observations. (There was no child whose consecutive weekly specimen tested positive for *S. pneumoniae*). Used MS Excel™ (Microsoft Office 2010) and Epi Info v3.5 (CDC, Atlanta, GA, USA) for data entry. Laboratory and clinical datasets were merged and analyzed using Stata 12SE (Stata Corp, College station, Texas, USA). To calculate adjusted odds ratios, combined AURI and ALRI cases and used logistic regression to compare pneumococcal detection in specimens collected from symptomatic (AURI/ALRI) children compared to asymptomatic children, adjusting for age group, sex, month of detection and co-detection of any other bacterial and viral pathogens.

### Ethical clearance

The study was approved by the Institutional Ethics Committee (IEC) of AIIMS, New Delhi, India. The institutional review board of the US Centers for Disease Control and Prevention relied on the review of the IEC of AIIMS. Written informed consent was obtained from all parents or guardians of enrolled children and assent was obtained from children aged ≥7 years.

## Results

Study followed 3765 children weekly for two years and collected a total of 4348 upper respiratory specimens. During the two years, 3328 specimens collected from children with ARI (including 836 ALRI and 2492 AURI) and 1020 specimen from asymptomatic children (Fig 1). There were 162 specimens from 131 children which were collected within 14 days of last sample collection. Overall, of the 4348 specimens tested, and detected *S. pneumoniae* in 220 (5.1%, 95% CI: 4.4–5.8) specimens, including 186 (5.6%, 95% CI: 4.8–6.4) of 3328 specimens from ARI cases (n = 3328) as shown in Table 1. Of these detections among ARI specimens, 118 (63.4%) were from AURI and 68 (36.6%) from ALRI. *S. pneumoniae* was detected more frequently among children <1 year of age compared to older children, both among symptomatic children (25.0%; 95% CI: 21.4–28.9 ≤1 year, versus 1.9%; 95% CI 1.4–2.4 >1 year) and asymptomatic children (15.2%; 95% CI: 10.2–21.5 ≤1 year, versus 0.9%, 95% CI 0.4–1.8 >1 year). Further, of 160 detections among infants aged <1 year, 51 (41 with ARI and 10 asymptomatic) were among infants aged <6 months; the youngest age of carriage was 36 days which was in an asymptomatic infant. There was no significant difference in *S. pneumoniae* detection between boys and girls (S2 Table).

Among the children with *S. pneumoniae* detections, also detected *Haemophilus influenzae* type b (17, 7.7%), and respiratory viruses including rhinovirus (30, 13.6%), RSV (10, 4.5%), influenza (5, 2.3%), parainfluenza viruses (3, 1.4%), and metapneumovirus (1, 0.5%).

Using asymptomatic children as the reference group, found statistically significant association between *S. pneumoniae* detection among children with AURI (adjusted OR: 2.6; 95%CI: 1.5–4.4; p = 0.000), and ALRI (adjusted OR: 2.0; 95% CI: 1.1–3.5; p = 0.116), when adjusted for age groups, sex, co-detection of other bacterial or respiratory viral pathogen and month of detection (S2 Table).

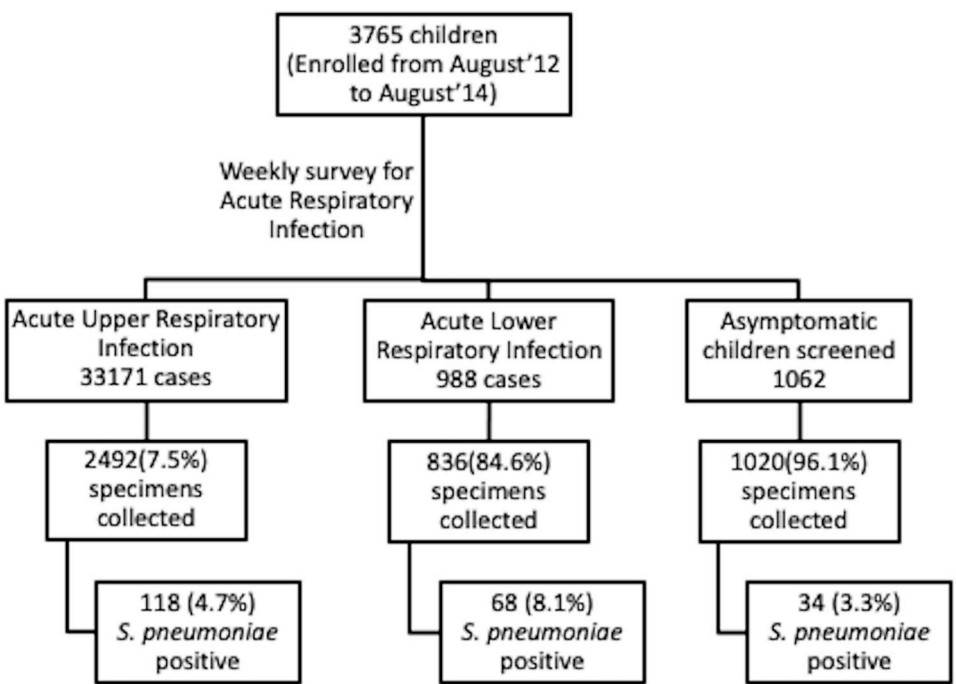

**Fig 1. Flow-chart of specimens (nasal/throat swab) collection in children less than ten years from a rural community of north India.**

Identified 42 diverse pneumococcal serotypes among this rural cohort of children aged less than 10 years. The ten most frequent serotypes were 6B/C, 19A, 19F, 23F, 35A/B/C, 15B, 14, 9V, 11A/C/D & 34/47; Combined, these accounted for 55% of the isolates (S1 Table). Ten isolates (4.5%) were non-typeable. Serotypes included in the currently used pneumococcal vaccines (PCV-10, PCV-13 and Pneumococcal polysaccharide vaccine (PPV)-23) are shown in Fig 2. We found that 45% of isolates had serotypes included in PCV-13, commonly used vaccine in India. The serotypes 35A/B/C, 15B, and 34/47, not included in PCV-13, were frequently detected among children with AURI and ALRI but not among asymptomatic children in this community.

Among 220 pneumococcal isolates, susceptibility was high to penicillin (90%; 95% CI: 85.0–94.0%), ceftriaxone (100%; 95% CI: 98.3–100%), vancomycin (99.5%; 95% CI:97.5–99.9),

**Table 1. Isolation of *S. pneumoniae* from nasal/throat swabs in children aged less than 10 years by age and symptom status in a rural community of north India from 2012–14 (N = 4348 specimens).**

| | Asymptomatic children | | | Acute Respiratory Infection | | |
|---|---|---|---|---|---|---|
| | No. of specimens tested | *S. pneumoniae* | | No. of specimens tested | *S. pneumoniae* | |
| | | N | % (95% CI) | | N | % (95% CI) |
| 0 to <1 year | 171 | 26 | 15.2 (10.2–21.5) | 536 | 134 | 25 (21.4–28.9) |
| 1 to <2 years | 320 | 6 | 1.9 (0.7–4.0) | 602 | 16 | 2.7 (1.5–4.3) |
| 2 to <5 years | 488 | 2 | 0.4 (0–1.5) | 1155 | 17 | 1.5 (0.9–2.3) |
| 5 to <10 years | 41 | 0 | 0.0 (0–8.6)* | 1035 | 19 | 1.8 (1.1–2.9) |
| **Total** | **1020** | **34** | **3.3 (2.3–4.6)** | **3328** | **186** | **5.6 (4.8–6.4)** |

* One sided 97.5% Confidence Interval (CI)

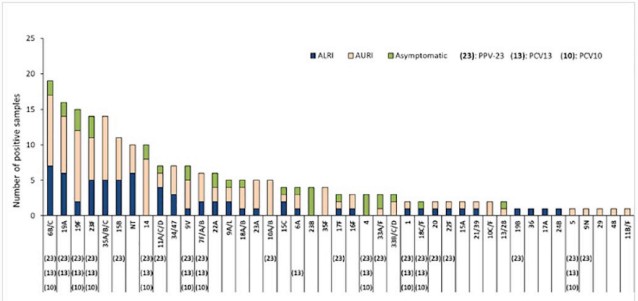

**Fig 2. Detection of serotypes of *Streptococcus pneumoniae* from upper respiratory tract of children aged <10 years in a rural community of northern India from 2012–14 (n = 220).** ALRI: Acute Lower Respiratory Tract Infection, AURI: Acute Upper Respiratory Tract Infection, PPV: Pneumococcal polysaccharide vaccine, PCV: Pneumococcal conjugate vaccine.

levofloxacin (97.7%; 95% CI 94.7–99.2%), chloramphenicol (97.3%; 95% CI 94.2–99.0), and erythromycin(90.9%; 95% CI 86.3–94.3%), but low to co-trimoxazole (6.4%; 95% CI 3.5–10.4%). There were nine (4%; 95% CI 1.9–7.6%) pneumococcal isolates that were multidrug-resistant to three antibiotics, all of which were resistant to co-trimoxazole. The multi-drug resistant serotypes were 7 F/A/B (n = 2), NT (n = 2), 11A/C/D, 19F, 33B/C/D, 9A/L, 9N. The most common penicillin resistant serotypes were 14 (n = 4), NT (n = 4), followed by 15B, 19F, 23A and 11A/C/D each with two isolates and single isolate with serotype 6B/C, 9N, 10A/B, 11B/F, 13/28, 23A and 33B/C/D. Minimum inhibitory concentration ranges (0.125 to 1.5 μg/ml) for penicillin non-susceptible isolates. Twenty out of 220 pneumococcal isolates were erythromycin resistant, and most common serotypes were 7F/A/B (n = 6), 23F (n = 4) and 19F (n = 2).

## Discussion

The present study was carried out in a north Indian rural community, to determine the carriage and serotypes of *S. pneumoniae* isolated from the upper respiratory tracts of children with ARI and asymptomatic children. Colonization with *S. pneumoniae* is considered a first step to pneumococcal disease and nasopharyngeal carriage has been used as a surrogate for circulating invasive strains in communities [22]. Similar to other studies, we found high pneumococcal carriage among infants (<1 yr) [23–25], with the earliest detection in a 36-days old infant. We found low overall carriage rates among children aged ≥1year in this community. These finding might be explained by the high prevalence of antibiotic use, as seen in some studies in rural northern India [26], or acquisition of cumulative immunity from multiple exposures to various serotypes [27].

Prevalent serotypes in our study are different from those identified previously in this region or in other parts of India. Five most frequent serotypes in current study were 6B/C, 19A, 19F, 23F and 35A/B/C representing >35% of serotypes found in this area during the study period. A study from the region of northern India in 2007, children aged <3 years a pediatrics outpatient clinic showed the pneumococcal carriage rate was 6.5% and frequent serogroups/types were 1, 6, 14, and 19 [28]. While studies from southern India showed the most frequent serogroups/types among children were 19, 6, 15, 23, 9, 35 and 10 [23, 24], and children from the eastern part of India showed different serogroups/types, 33, 8, 1, 19, 6 and 23 [25]. Neighbouring countries like Bangladesh, Pakistan, Sri Lanka and Nepal had some common circulating serotypes: 1, 5, 14, 19F, 19A, 23F and 6B [9]. The frequently observed serotypes, such as 14, 1,

5, 19F and 6B in some of the recent studies from India on invasive pneumococcal disease among children <5 years old, were also detected in our study [7–9].

The antimicrobial susceptibility pattern of *S. pneumoniae* isolated in this cohort is similar to that found in previous studies from India and elsewhere [29]. Antimicrobial resistance was most common against co-trimoxazole, similar to the findings in previous studies from India [13, 24, 25]. Asian Network for Surveillance of Resistant Pathogens (ANSORP) multi-countries collaborative work reported that pneumococcal isolation from India had the lowest resistant to penicillin. A recent study [7] showed slightly increase in the number of penicillin non-susceptible *S. pneumoniae* with 8% (n = 29). Our study showed high susceptibility to penicillin, as found elsewhere in India [25], and unlike the high penicillin resistance observed in other Asian countries like Thailand, Singapore, Hong-Kong had high resistance to penicillin 68%, 44% and 49%, respectively [9, 30]. In a global review, high rates of resistance against both penicillin and erythromycin were reported in 6B, 6A, 9V, 14, 15A, 19F, 19A, and 23F [31]; however, in current study serotype 19F showed resistance against both the antibiotics. Two non-vaccine serotypes 35A/B/C and 11A/C/D, found in this rural community and previously have been shown to be associated with biofilm formation [32], which may be responsible for the development of antibiotic resistance.

In this study, 45% of isolates were vaccine serotypes, while the remaining isolates were either non-typable (10%) or non-vaccine type. The Government of India approved the introduction of pneumococcal vaccine into the universal immunization programme (UIP) in 2017, starting with high-burden states [5]. Our study area is not considered a high-burden area and pneumococcal vaccine was not available in the area during this study. Therefore, our study documents baseline rates of pneumococcal detection and circulating serotypes that can be used to monitor the impact of pneumococcal vaccine introduction in the future. In other settings, the introduction of mass immunizations has resulted in replacement of non-vaccine serotypes in the community and reduction in the prevalence of drug resistant serotypes [31]. However, it is still unclear whether the replacement of serotypes is associated with invasive pneumococcal disease.

One of the limitations of this study was collection of only upper respiratory tract specimens for detection of *S. pneumoniae*, which makes it difficult to attribute any etiological role in ARIs. This study was not originally designed to assess pneumococcal carriage. It leveraged a community-based surveillance platform for viral pathogens among children [16]. Despite these limitations, this study is one of few community-based pneumococcal carriage studies among children in India and provides insights into the dynamics of circulating pneumococcal serotypes in children 10 years of age and younger in rural India prior to the introduction of pneumococcal vaccine.

Findings of current study can inform the national guidelines on antimicrobial use for pneumococcal disease and provide critical information about baseline rates of *S. pneumoniae* carriage and circulating serotypes that can be used to measure the impact of vaccine introduction if the national vaccination program is expanded. The difference in circulating serotypes seen between our study and previous carriage studies suggests that such community-based carriage studies can be used to better understand the prevalence of different pneumococcal serotypes in India and the impact of vaccination.

## Supporting information

**S1 Data.**
(XLSX)

**S1 Table. Pneumococcal serotypes distribution among 220 isolates from children in rural northern India, 2012–2014.**
(PDF)

**S2 Table. Multi-variate analysis of factors associated with detection of pneumococcus in the upper respiratory specimens collected from children aged <10 years in rural north India.** AURI: Acute upper respiratory infection, ALRI: Acute lower respiratory infection, RSV: *Respiratory syncytial virus*, HRV: *Human rhinovirus*, HMPV: *Human metapneumovirus*, PIV: *Parainfluenza virus*, Hib: *Haemophilus influenzae type b*.
(PDF)

## Acknowledgments

We are indebted to the villagers of the study area for their support and patience despite repeated visits during the study. We are thankful to the study participants and the project staff for their cooperation and support for this study. We express our profound gratitude to Dr Renu Lal and Marc-Alain Widdowson from CDC, and Prof Shobha Broor from AIIMS, who conceptualized this surveillance platform used for this study. We would like to express our special thanks to Dr Srinivas Acharya Nanduri from Division of Bacterial Diseases, CDC for his comments and suggestions for the manuscript.

**Disclaimer:** The findings and conclusions in this report are those of the authors and do not necessarily represent the official position of CDC.

Conferences, where this study has been presented:

1stIMRP (International meeting on respiratory pathogens), 2–4 September 2015, Singapore (Abstract no.-ASN0-211)

9th International symposium on pneumococci and pneumococcal diseases (ISPPD), 9–13 March 2014, Hyderabad, India (Abstract no. -ISPPD-0268)

## Author Contributions

**Conceptualization:** Sambuddha Kumar, Arti Kapil, Siddhartha Saha, Bimal Kumar Das, Seema Jain, Anand Krishnan.

**Data curation:** Sambuddha Kumar, Debjani Ram Purakayastha, Arti Kapil, Siddhartha Saha, Ritvik Amarchand, Rakesh Kumar.

**Formal analysis:** Sambuddha Kumar, Siddhartha Saha, Ritvik Amarchand, Rakesh Kumar.

**Funding acquisition:** Siddhartha Saha, Kathryn E. Lafond, Seema Jain, Anand Krishnan.

**Investigation:** Sambuddha Kumar, Debjani Ram Purakayastha, Arti Kapil, Bimal Kumar Das, Ritvik Amarchand, Rakesh Kumar, Anand Krishnan.

**Methodology:** Sambuddha Kumar, Debjani Ram Purakayastha, Arti Kapil.

**Project administration:** Debjani Ram Purakayastha, Arti Kapil, Siddhartha Saha, Fatimah S. Dawood, Bimal Kumar Das, Ritvik Amarchand, Kathryn E. Lafond, Seema Jain, Anand Krishnan.

**Resources:** Debjani Ram Purakayastha, Arti Kapil, Bimal Kumar Das, Ritvik Amarchand, Rakesh Kumar, Anand Krishnan.

**Software:** Rakesh Kumar.

**Supervision:** Arti Kapil, Siddhartha Saha, Fatimah S. Dawood, Bimal Kumar Das, Kathryn E. Lafond, Seema Jain, Anand Krishnan.

**Validation:** Ritvik Amarchand, Anand Krishnan.

**Visualization:** Arti Kapil, Bimal Kumar Das, Kathryn E. Lafond, Seema Jain, Anand Krishnan.

**Writing – original draft:** Sambuddha Kumar, Arti Kapil, Bimal Kumar Das.

**Writing – review & editing:** Sambuddha Kumar, Debjani Ram Purakayastha, Arti Kapil, Siddhartha Saha, Fatimah S. Dawood, Bimal Kumar Das, Ritvik Amarchand, Rakesh Kumar, Kathryn E. Lafond, Seema Jain, Anand Krishnan.

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
