## [Decision Letter · Decision Letter 0]

29 Sep 2020

PONE-D-20-25601

Carriage rates and antimicrobial sensitivity of pneumococci in the upper respiratory tract of children less than ten years old, in a north Indian rural community

PLOS ONE

Dear Dr.Bimal Kumar Das 

Thank you for submitting your manuscript to PLOS ONE. After careful consideration, we feel that it has merit but does not fully meet PLOS ONE’s publication criteria as it currently stands. Therefore, we invite you to submit a revised version of the manuscript that addresses the points raised during the review process.

We look forward to receiving your revised manuscript.

Kind regards,

Silvia Ricci

Academic Editor

PLOS ONE

Journal Requirements:

2.In your Data Availability statement, you have not specified where the minimal data set underlying the results described in your manuscript can be found. PLOS defines a study's minimal data set as the underlying data used to reach the conclusions drawn in the manuscript and any additional data required to replicate the reported study findings in their entirety. All PLOS journals require that the minimal data set be made fully available. For more information about our data policy, please see http://journals.plos.org/plosone/s/data-availability.

Reviewers' comments:

Reviewer's Responses to Questions

**Comments to the Author**

1. Is the manuscript technically sound, and do the data support the conclusions?

Reviewer #1: Yes

2. Has the statistical analysis been performed appropriately and rigorously? 

Reviewer #1: Yes

3. Have the authors made all data underlying the findings in their manuscript fully available?

Reviewer #1: Yes

4. Is the manuscript presented in an intelligible fashion and written in standard English?

Reviewer #1: Yes

5. Review Comments to the Author

Reviewer #1: The title: Carriage rates and antimicrobial sensitivity of pneumococci in the upper respiratory tract

2 of children less than ten years old, in a north Indian rural community

This topic is an important topic indicating pneumococcal resistance in children less than ten years with high risk to moderate risk of pneumococcal infections and mostly with high resistance rates for rural areas. The manuscript should have the following points for corrections

reviewer points:

1. Follow the author instructions regarding abstract writing, it should be in one paragraph

2. Line 28: anti-microbial, no need for the dash

3. again in the abstract methods, it is stated that you took only nasal swabs from children less than one year, but the rest children both nasal and throat, but in the results (nasal/throat for both), please explain

4. lines 34, 35 and 36: (with 6B/C, 19A, 19F, 23F, 35A/B/C, 15B, 14 and 11A/C/D

35 accounting for 50%). You should indicate the exact percentage of each serotype from the highest to the lowest, showing the importance of each and give the reader an idea about the coverage of the pneumococcal conjugate vaccines, and this may also indicate or show the importance of the future PCVs PCV15 or PCV20.

5. Line 37 in the abstract, it says less than 10% penicillin intermediate resistance: Please indicate the exact resistance rates and also high grade resistance for the penicillin

6. The introduction is mainly talking about India and is short, it needs extension and talk more internationally, especially in countries similar to India

7. line 66, same anti-microbial, no need for the dash

8. The study is about Low and Upper RI, were there any pneumonia cases for the children diagnosed?

9. In line 79. change the sentence starting with we, since it is not scientifically correct

10. lines 91-93: Is there any importance of including this statement?

11. Line 103: (for testing for), here English revision

12. Line 104: be sure to make all CO2 written the same in the manuscript (the number 2 should be low script)

13. make sure there are no double spaces between words in all of the manuscript, look at line 119.

14. line 127, the reference was put after the point (wrong)

15. lines 124-130: You have mentioned in the abstract penicillin intermediate resistance and even in line 199 about penicillin resistance, but penicillin was not mentioned in the methods in this paragraph

16. line 132: What is s/he?

17. I do not encourage you to write we analyzed, we considered, we used, we also detected, We identified...etc. This is not preferable in the manuscripts

18. lines 153-154: This included 162 specimens collected within 14 days of last specimen collection from the same child. Please clarify 162 samples from the same child?

19. Indicate the serotypes in the figure with PCV serotypes or not

6. PLOS authors have the option to publish the peer review history of their article (what does this mean?). If published, this will include your full peer review and any attached files.

Reviewer #1: No

---

## [Author Response · Author response to Decision Letter 0]

6 Dec 2020

Dear Dr. Silvia Ricci

Academic Editor, PLOS ONE.

Thanks for giving chance to submit a revised version of the manuscript that you have addressed the points raised during the review process. Thank you so much for review this manuscript such a sensible way and reasonable comments.

---

## [Decision Letter · Decision Letter 1]

21 Jan 2021

Carriage rates and antimicrobial sensitivity of pneumococci in the upper respiratory tract of children less than ten years old, in a north Indian rural community

PONE-D-20-25601R1

Dear Dr. Bimal Kumar Das, 

We’re pleased to inform you that your manuscript has been judged scientifically suitable for publication and will be formally accepted for publication once it meets all outstanding technical requirements.

Kind regards,

Silvia Ricci

Academic Editor

PLOS ONE

Additional Editor Comments (optional):

Reviewers' comments:

Reviewer's Responses to Questions

**Comments to the Author**

1. If the authors have adequately addressed your comments raised in a previous round of review and you feel that this manuscript is now acceptable for publication, you may indicate that here to bypass the “Comments to the Author” section, enter your conflict of interest statement in the “Confidential to Editor” section, and submit your "Accept" recommendation.

Reviewer #1: All comments have been addressed

2. Is the manuscript technically sound, and do the data support the conclusions?

Reviewer #1: Yes

3. Has the statistical analysis been performed appropriately and rigorously? 

Reviewer #1: Yes

4. Have the authors made all data underlying the findings in their manuscript fully available?

Reviewer #1: Yes

5. Is the manuscript presented in an intelligible fashion and written in standard English?

Reviewer #1: Yes

6. Review Comments to the Author

Reviewer #1: The reviewer have checked all the points raised and were answered and the responses meet formatting specifications.

7. PLOS authors have the option to publish the peer review history of their article (what does this mean?). If published, this will include your full peer review and any attached files.

Reviewer #1: No

---

## [Editor Report · Acceptance letter]

25 Jan 2021

PONE-D-20-25601R1 

Carriage rates and antimicrobial sensitivity of pneumococci in the upper respiratory tract of children less than ten years old, in a north Indian rural community 

Dear Dr. Das:

I'm pleased to inform you that your manuscript has been deemed suitable for publication in PLOS ONE. Congratulations! Your manuscript is now with our production department. 

Kind regards, 

on behalf of

Dr. Silvia Ricci 

Academic Editor

PLOS ONE